# Public Health Regulations and Policies Dealing with Preparedness and Emergency Management: The Experience of the COVID-19 Pandemic in Italy

**DOI:** 10.3390/ijerph19031091

**Published:** 2022-01-19

**Authors:** Luna Aristei, Floriana D’Ambrosio, Leonardo Villani, Maria Francesca Rossi, Alessandra Daniele, Carlotta Amantea, Gianfranco Damiani, Patrizia Laurenti, Walter Ricciardi, Maria Rosaria Gualano, Umberto Moscato

**Affiliations:** 1Department of Law, LUISS Guido Carli University, 00198 Rome, Italy; laristei@luiss.it; 2Section of Hygiene, University Department of Life Sciences and Public Health, Università Cattolica del Sacro Cuore, 00168 Rome, Italy; florianadambrosio@libero.it (F.D.); gianfranco.damiani@unicatt.it (G.D.); patrizia.laurenti@unicatt.it (P.L.); walter.ricciardi@unicatt.it (W.R.); umberto.moscato@unicatt.it (U.M.); 3Section of Occupational Health, Department of Woman and Child Health and Public Health, Università Cattolica del Sacro Cuore, 00168 Rome, Italy; mariafrancesca.rossi01@icatt.it (M.F.R.); alessandra.daniele02@icatt.it (A.D.); carlotta.amantea01@icatt.it (C.A.); 4Department of Woman and Child Health and Public Health, Fondazione Policlinico Universitario A. Gemelli, IRCCS, 00168 Rome, Italy; 5Department of Public Health Sciences, University of Turin, 10124 Turin, Italy; mariarosaria.gualano@unito.it; 6Section of Occupational Health, Department of Woman and Child Health and Public Health, Fondazione Policlinico Universitario A. Gemelli, IRCCS, 00168 Rome, Italy

**Keywords:** health emergency, management, disaster legislation, preparedness, COVID-19

## Abstract

Worldwide, the management of health emergencies requires a high degree of preparedness and resilience on the part of governments and health systems. Indeed, disasters are becoming increasingly common, with significant health, social, and economic impacts. Living in a globalized world also means that emergencies that occur in one country often have an international, in some cases global, spread: the COVID-19 pandemic is a cogent example. The key elements in emergency management are central governance, coordination, investment of resources before the emergency occurs, and preparedness to deal with it at all levels. However, several factors might condition the response to the emergency, highlighting, as for Italy, strengths and weaknesses. In this context, policies and regulation of actions to be implemented at international and national level must be up-to-date, clear, transparent and, above all, feasible and implementable. Likewise, the allocation of resources to develop adequate preparedness plans is critical. Due to COVID-19 pandemic, the European Commission proposed the temporary recovery instrument NextGenerationEU, as well as a targeted reinforcement of the European Union’s long-term budget for the period 2021–2027. The pandemic highlighted that it is necessary to interrupt the continuous defunding of the health sector, allocating funds especially in prevention, training and information activities: indeed, a greater and more aware public attention on health risks and on the impacts of emergencies can help to promote virtuous changes, sharing contents and information that act as a guide for the population.

## 1. Introduction

The management of disasters and emergencies is a core function of Public Health. Indeed, natural or humanmade extraordinary events might constitute a public health risk that require a timely, coordinated, and efficient response from governments and healthcare systems. Disasters can be natural (geophysical, meteorological, hydrological, climatological, biological, extraterrestrial) or technological (humanmade—intentional or nonintentional) [1], and they are defined as any occurrence that causes damage, ecological disruption, and loss of human life, resulting in a serious failure of the functioning of the community [2]. Moreover, the impact of these situations often exceeds the response capabilities of the community, requiring external assistance (at the national or international level) in terms of human, economic, structural and instrumental resources [3,4]. In recent years, the number of disasters and public health emergencies increased in frequency and intensity, causing a massive social disruption and thus requiring specialized management [5,6]. In particular, it is estimated that between 2005 and 2015, over 700,000 people died as a result of these phenomena, more than 1.4 million have been injured and 23 million have lost their homes, while economic losses exceeded 1.3 trillion USD. In addition, natural disasters between 2008 and 2012 caused the migration of about 144 million people [6]. Moreover, only in 2019, natural disasters involved about 95 million people with 11,775 deaths and 103 billion USD in economic losses worldwide [3].

Of note, an important issue for Public Health at the international level is represented by microbiological emergencies, which are becoming increasingly frequent. Indeed, the World Health Organization (WHO) monitored worldwide 1483 epidemics in 172 Countries between 2011 and 2018 [7] and, of these, epidemic-prone diseases such as influenza, severe acute respiratory syndrome (SARS), Middle East respiratory syndrome (MERS), Ebola, Zika, and SARS-CoV-2 represent a serious risk for the onset of epidemics or pandemics, as they are characterized by a potentially fast-spreading outbreaks worldwide [7]. Therefore, disasters have a huge health, social, and economic burden, with direct and indirect long-term consequences on the population, such as psychological and behavioral effects (increase of depression, anxiety, insomnia, stress) [5,8,9], economic losses [10,11] and social repercussions (increase of poverty, migrations of entire populations) [10,12,13]. In this context, adequate emergency preparedness and resilience of governments and healthcare systems, developed through a coordinated and integrated response guided by strong leadership and based on international policies, can lead to significant savings in lives and in economic resources [14]. Preparedness, in fact, is the ability to effectively anticipate, respond to, and recover from public health emergencies through vision, knowledge, skills of planning and organization of governments, communities, and individuals, at local, regional, national, and international level [15]. In particular, seven main skills (planning, coordination, timely diagnosis, evaluation, investigation, response, and communication) are required to guide strategies and mechanisms to face public health emergencies. Thus, these skills should be applied to four priority areas of action that can be implemented at both the national (and local) and supranational (and global) levels: understanding risk, enhancing governance, investing in risk reduction for resilience, and improving disaster preparedness in recovery and reconstruction [6]. Developing adequate preparedness plans, therefore, is a global priority, through transparent and coordinated policies shared by all countries.

In this commentary, we present an overview of the policies, regulatory frameworks and legislation on health emergency management at global and European level. Then we focus on the Italian COVID-19 pandemic as an example of management of health emergencies. Finally, this paper concludes by proposing some directives about the management of future emergencies. 

## 2. Legislation and Policies on Health Emergency Management

### 2.1. Global and European Level

The attempt to manage and coordinate the response to health emergencies at international level began in 19th century. The first International Health Conference, in fact, was organized in 1851 after the European cholera epidemy (1830–1847) [16]. Then, the first two international conventions on health emergencies were approved in 1892 and in 1897 on cholera and plague control, respectively [17]. In 1946, the WHO was created and in 1969 the International Health Regulations (IHRs), an instrument of international law, were approved to share epidemiological information to prevent, respond and control the spread of infectious diseases across borders, without interfering with international trade and movement [18]. In May 2005, due to globalization, IHRs were adjourned, becoming legally binding. In this context, IHRs defined the notification criteria to WHO for infectious diseases of urgent importance for international public health, such as those with rapid transmission, high lethality, newly identified syndrome, and possible restrictions on trade or travel. Moreover, Member States had the duty to develop, strengthen and maintain the capacity to respond promptly and effectively to public health risks and health emergencies of international concern [19].

In Europe, the first emergency regulation was approved in 1998 and Council Decision n. 2001/792 established the European Civil Protection, (reformed in 2013) to strengthen and coordinate disaster prevention, preparedness, and response [20,21]. Then, two action plans were adopted in 2005 (n. 605 and 607) to help states draw up and adopt health management plans [22,23]. Additionally, national, European (European Centre for Disease Prevention and Control—ECDC) and global (WHO) management and control centers of public health emergency were linked with each other to promptly notify and activate alert situations. In March 2004, the European Commission adopted a preparedness plan COM(2004)201 that sets out actions in management and coordination, surveillance, prevention, mitigation and response, communication, civil protection and research’s areas [24]. It was then amended by Communication COM(2005)607 [23], which identified six phases of an influenza pandemic as defined by the WHO (no virus subtype circulating; circulation of a virus subtype between animals with risk to humans; human infection with no interhuman transmission; limited interhuman transmission; increased but localized interhuman transmission; increasing and sustained transmission among the population) [25]. In each phase, responsibilities are shared between the European Commission, Member States and the ECDC (established in 2004). Decision n. 1082/2013 repealed 1998 decision, establishing that Member States and the Commission consult each other within the Health Security Committee (HSC), through the creation of the Early Warning and Response System (EWRS) to notify serious cross-border threats to health [21]. Once a risk alert has been notified, the Commission provides the National Authorities and the HSC a risk assessment of the threat’s potential severity to public health, coordinating actions. 

During the COVID-19 pandemic, the European Commission adopted Communication COM(2020)724 to strengthen European Union (EU)’s resilience and coordination to cross-border health threats [26]. This Communication aims to strengthen the ECDC’s mandate and expand that of the European Medical Agency (EMA). Additionally, this Communication strengthens emergency management tools (such as countermeasures or medical devices) when national capacities are insufficient, complementing EU Civil Protection capabilities. On 16 September 2021, the European Commission inaugurated the Health Emergency preparedness and Response Authority (HERA) to prevent, detect, and respond rapidly to health emergencies. In particular, if a public health emergency is declared at EU level, HERA can move quickly to emergency operations activating emergency funding and initiating monitoring mechanisms. 

### 2.2. Financial Interventions as COVID-19 Emergency Response in Europe

On 2 May 2018, the European Commission presented its proposal for the next long-term EU budget and on 27 May 2020, due to COVID-19, it proposed the temporary recovery instrument NextGenerationEU (NGEU) [27], as well as a targeted reinforcement of the EU’s long-term budget for the period 2021–2027. On December 2020, the EU Council adopted the long-term budget 2021–2027 [28] and the European Parliament and the Council reached an agreement on the Recovery and Resilience Facility (with EUR 723.8 billion in loans and grants), the key instrument underpinning NGEU [29]. NGEU is a temporary recovery facility (more than EUR 800 billion) that will help repair the immediate economic and social damage caused by COVID-19 to create a greener, more digital, and resilient Europe. It will also allocate additional funding to other European programs or funds, such as Horizon Europe or InvestEU. Moreover, there are other plans such as REACT-EU that allocates EUR 50.6 billion for the expansion of crisis response measures and EU4Health, to address the resilience of health systems (EUR 5.3 billion) [30]. Established by Regulation (EU) 2021/522, EU4Health will improve and promote health in the EU, addressing cross-border health threats. HERA’s activities will also have a budget of EUR 6 billion from the current Multiannual Financial Framework for the period 2022–2027. 

### 2.3. Italian Legislation on Health Emergency Management

The Italian Constitution does not contain specific rules on the state of emergency. However, based on article 77, the Government can adopt decrees in the cases of extraordinary necessity and urgency, concerning the concrete management of emergencies; also, article 120 allows the Government to replace local authorities in events of danger to public safety and security. Moreover, Article 117(3) states that the state must establish health principles. 

Law 8 December 1970, No. 996, was the first identifying ordinary (Interior Minister, Prefect, Regional Government Commissioner, Mayor) and extraordinary (Extraordinary Commissioner) Civil Protection’s bodies as well as their competences [31]. In the Civil Protection’s system, state bodies (Prefect and Government Commissioner) mainly managed the emergency. Law 24 February 1992, No. 225, established the National Service of Civil Protection that deals with relief, forecasting, and prevention, defining natural disasters’ causes and the risks for the territory. It also takes all the appropriate and necessary actions to reduce or avoid damages from natural disasters. The 1992 law reorganized Civil Protection’s structure as a coordinated system of competences in which public and private entities participate. Law 24 February 1992 was repealed by legislative decree 2 January 2018, No. 1 (Civil Protection Code) [32]. This code reiterates a polycentric model, in which all local and voluntary bodies are involved, providing a coordinated emergency management system (relief and assistance interventions, allocation of funds).

Indeed, in case of emergency, a bottom-up pyramidal response mechanism is activated, starting from the level closest to citizens. Therefore, the mayor (law 18 August 2000, n. 267) directs and coordinates relief operations, assisting the population and organizing municipal resources [33]. He/she can also approve contingent and urgent orders to prevent and eliminate serious dangers threatening public safety and urban security. The mayor has then to draw up the Municipal Emergency Plan (MEP) and if he/she is unable to cope with the event by its own means, the higher local levels are involved (Province, Prefecture, Region, Ministry of Health—State). Moreover, at local level, in the case of infectious emergency, the Prevention Department (a technical-functional structure of the Local Health Authorities) are responsible for the management of infectious cases and outbreaks. 

Nationally, the Council of Ministers’ President coordinates measures with Civil Protection Department, appoints delegated Commissioners and specific task forces, and issues emergency ordinances to avoid dangerous or damaging situations. The Council of Ministers, based on President’s proposal, decrees the state of emergency indicating its territorial extension and duration (it cannot exceed 12 months, and can be extended for no more than a further 12 months) [32]. 

Finally, Agencies in support of the Ministry of Health carry out technical-scientific consulting activities (National Health Institute—ISS), drug regulatory activities (Italian Medicines Agency—AIFA), and health performance monitoring and control (National Agency for Regional Health Systems—AGENAS).

### 2.4. Financial Interventions as COVID-19 Emergency Response in Italy

The Italian Recovery and Resilience Plan was approved on 22 June 2021 by the European Commission [34]. It responds to the urgent need to foster a strong recovery, making Italy more sustainable, resilient, and better prepared. The Plan will be supported by EUR 68.9 billion in grants and EUR 122.6 billion in loans, and all reforms and investments must be implemented by August 2026. Mission 6 directs resources to resilience’s strengthening and timeliness of the National Health System’s response to emerging infectious diseases with high morbidity and mortality, as well as to other health emergencies. Additionally, it tends to develop proximity healthcare and stronger integrations between health, social, and environmental policies to foster effective social inclusion. It also aims to invest in medical assistance’s digitalization, promoting the spread of the Electronic Health Record and telemedicine, adopting digital technologies in the field of medical assistance and prevention services. Regarding territorial and proximity medicine, the investments of the Plan are oriented toward strengthening the instruments for care in the territory and in the homes of patients (e.g., using telemedicine), especially those with chronic diseases, to leave hospital care only when necessary. An example is represented by the creation of “Case della Comunità–Community Homes”, which are health care facilities that promote a multidisciplinary intervention model for planning social interventions and social-health integration. 

To fight the pandemic, Italy also approved several decree laws for urgent measures that have provided for a significant increase finance in the standard national health requirement (+EUR 1410 million for 2020 established by the so-called Cura Italia decree) [35]. Cura Italia also recognized to Regions, for the whole state of health emergency period, to issue special insurance coverage for the purchase of goods related to the management of the epidemiological crisis. Then, Liquidity Decree provided a tax credit for companies to sanitize workplaces, purchase surgical masks and personal protective equipment, extended by Cura Italia also to non-commercial entities [36]. In addition, through the Fund for national emergencies (refinanced by the Rilancio Decree) [37] Cura Italia financed the purchase of facilities and equipment specifically for the treatment of COVID-19 patients, finances used mainly by the Civil Protection Department and the Extraordinary Commissioner for the emergency.

## 3. The Management of the COVID-19 Pandemic in Italy: Strengths and Weaknesses

Italy was the first western country affected by the COVID-19, representing the frontline against the pandemic, with a mortality rate among the highest in Europe, especially during the first wave [38]. In this context, in order to quickly respond and manage the emergency, the government declared the state of emergency on 31 January 2021 [39]. Therefore, several immediate actions have been implemented to contain the spread of the virus, also in accordance with the eight pillars proposed by WHO [40]. Among these, the creation of specific task forces of experts and the involvement of technical and scientific support bodies at national (ISS, AIFA) and regional level to provide scientific advice to the government, and the allocation of significant economic resources (EUR 3.7 billion in 2020 and EUR 1.7 billion in 2021) to health systems [41] in order to enhance epidemiological surveillance, testing capacity and laboratory activities, to increase hospital facilities and intensive care units beds, and to create of special units to manage COVID-19 patients, establishing a COVID-19 integrated surveillance system [41,42,43].

However, several factors have made it difficult to implement a rapid and coordinated response. In particular, Italy has 20 different regional health care systems [44], and decentralization may have hindered preparedness [45]. In fact, the weak coordination with regional bodies, has led, especially in the early phase of the pandemic, to a frail and uneven response, with some regions implementing autonomous policies (testing, contact tracing, containment measures) not always in line with the central government [43,44,46]. In addition, in some cases the lack of coordination between hospital and primary care and territorial services has resulted in an inefficient response, with saturation of hospitals and inability to manage patients [44,47]. Similarly, due to lack of previous economic efforts on digital innovation, the absence of a strong and implemented digital health structure has reduced the possibilities of telemedicine and home management of mild symptomatic cases [48].

Furthermore, Italy show several limits in capacity planning of the hospitals. In particular, the enormous pressure generated on the healthcare system during the first and the subsequent waves caused by variants (such as omicron) highlighted the need to develop models for hospital surge capacity planning [49,50,51]. Indeed, given the emergence of new waves, it is a priority to identify risk scenarios that consider many factors, before reaching the crisis point: the current status of the disease, how quickly it spreads (e.g., doubling time), the degree of containment measures being deployed, the availability of healthcare workers, the hospital capacity in terms of beds and ventilator requirements [52]. In this context, at the end of the first wave the task force of experts in collaboration with the main Italian technical-scientific bodies developed four risk scenarios still in use through the creation of specific indicators of probability (virus transmission capacity, time of doubling of cases, spread in working and school environments), impact on hospitals (occupancy of beds in ordinary wards or intensive care units) and resilience (degree of acceptance of hygiene, health and behavioral measures by the population, contact tracing capacity, ability to carry out early diagnosis and monitoring of positive cases) [53]. In this way, regions are classified in four areas—white, yellow, orange, and red—that correspond to as many risk scenarios and for which specific restrictive measures are foreseen [54]. The implementation of these systems has led to the conversion of hospitals into specific COVID-19 structures, highlighting a good degree of resilience of healthcare facilities. The identification of these parameters and risk scenarios allows the creation of models that can facilitate the reconfiguration for disaster-resilient health infrastructure, also applicable to other types of health emergencies (i.e., earthquakes, floods, other infectious outbreaks) [55,56].

Finally, the constant defunding of the National Health Service in Italy in recent decades, which has led to a shortage of healthcare workers, insufficient structures and technology and the absence of integrated management, explains, at least partly, the difficulties in the management of the first wave of the pandemic [41,44].

## 4. Conclusions

The COVID-19 pandemic must represent a moment of awareness and reflection in order to improve governance processes and the ability to respond and react to disaster events. The COVID-19 emergency has confirmed that it is necessary to work more on the analysis of local contexts in order to design targeted interventions at regional and national level, improving the interaction and coordination between different settings. Planning and, therefore proper organization, represent the key to adapt and direct decision-making processes. The pandemic has highlighted how the current governance, often still anchored to laws, regulations and bureaucratic systems or apparatuses belonging to historical periods and contexts completely different from the current or future ones, is only partially able to provide a timely and coordinated response. Moreover, system resilience is a key issue in managing healthcare emergencies. In fact, the adaptive capacity of hospitals and in general of all healthcare facilities is essential to ensure rapid adaptation to an emergency. The pandemic has led to the development of predictive models and methodologies for risk assessment and quantification on which response measures to the spread and containment of the virus are developed. These models should be implemented and made feasible, not only in organizational and managerial terms but also in relation to structural, environmental, plant, engineering and technological aspects that can also be adapted to the context of other health emergencies, to ensure the presence of a resilient system capable of providing a quick, specific and personalized response as much as possible, with a major impact in terms of lives saved and reduction of social and economic damage. To make these models feasible, however, coordination and collaboration between the structures are necessary, both at a territorial and regional level, and mainly, at a national and central level, to overcome the regulatory conflicts, technical and legislative regulations that still exist and are not sufficiently corrected and updated on the basis of the current pandemic experience (Table 1).

In this context, the new European agency HERA represents an important step forward, although it is limited to the European Union and it still insufficient to ensure a timely, transparent, and coordinated response. It is necessary, in fact, to provide an agency not only with prerogatives of direction and control, but also with effective powers of intervention that are, when necessary, substitutive of national failures in emergency management. A control room, therefore, of rapid health intervention and civil protection, with its own budget and operational decision-making, free from the veto of a single state, which can act quickly, with consistent and coordinated measures, with the ability to collect homogeneous data, with professionally trained and constantly updated staff, with a production capacity—its own or supply—of tested machinery and protective devices, vaccines, and specific drugs. Thus, not only during an emergency, but also in ordinary times, it is desirable to lean on a Preparedness European Agency to share of resources and data, with harmonization of public health and social measures for response. Such a network would assure and promote accountability by synthesizing, reviewing and assessing operative information and knowledge, and for critically evaluating the effects of public health decisions. Basically, also to try to overcome the gap, which currently exists, between the time when there is awareness of a critical emergency event and the unharmonized decisions to effectively prevent, halt, or delay the consequences of this event. Considering the Italian situation, despite the health regionalism, it is necessary to identify an adequate “chain of command”, able to exercise a leadership role to coordinate and integrate the skills of all institutions and actors involved, on whose collaboration lies the readiness to respond and the resilience of an integrated system. Finally, the pandemic has highlighted the need to interrupt the continuous defunding of the health sector, allocating funds especially in prevention, training and information activities: indeed, a greater and more aware public attention toward health risks and on the impacts of emergencies can help to promote virtuous changes, to share contents and information that act as a guide for the population.

## Figures and Tables

**Table 1 ijerph-19-01091-t001:** Insights to properly manage a health emergency in Europe.

Insights and Suggestions for Creating A Coordinated Health Emergency Management System in Europe
Establishment of a permanent European infrastructure with the capacity for rapid intervention in the event of a health emergency that may involve more than one European state, with autonomous management of its own budget and funds necessary for intervention and autonomy in operational decision-making, free from the veto of a single state.
Continuous funding and promotion of transnational collaboration, with central governance of emergency management and a task force for rapid local health intervention (“rapid intervention health task force” or RIHTF).
Creation or implementation of central European laboratories for the development of research and prevention, diagnostic and therapeutic methods to combat infectious and/or toxicological agents, or governance of the European network for the surveillance of communicable diseases (ECLDC).
Implementation of the real-time surveillance network, through digital tools and data interoperability, between States and Local Authorities.
Increasing the resilience of national and regional health systems through the development and adoption of predictive models and methodologies for risk assessment and quantification, and the study of harmonized decision-making processes that can be unambiguously adopted across EU countries.

## Data Availability

Not applicable.

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
