# Peer review of "Public Health Regulations and Policies Dealing with Preparedness and Emergency Management: The Experience of the COVID-19 Pandemic in Italy"

_ijerph, 2022, doi:10.3390/ijerph19031091_

Round 1
Reviewer 1 Report
The paper is interesting and addresses an important issue. while there are some concerns that should be addressed. first of all, I do believe the title of the paper is not appropriate. As a matter of fact, there are two issues in the title, first of all, "disaster" is very confusing in the title. as a matter of fact, there is a wide range of disasters which recently some of them particularly those are related to nature, are called hazard not disaster. However, this paper is focusing on Covid and pandemic. Therefore, I suggest the author should revise the title. In addition, the author discussed Italy in the body of the paper, maybe it is better to replace "European" with something that is related to Italy.
- But one of the main issues of this paper is hospitals and their capacities planning during covid have not been discussed significantly. While I followed NEWS and know about one year ago, many European countries have many problems in capacity planning of the hospitals and providing reliable services for patients. this is a very important issue that should be discussed. please see:
- Klein, M.G., Cheng, C.J., Lii, E., Mao, K., Mesbahi, H., Zhu, T., Muckstadt, J.A. and Hupert, N., 2020. COVID-19 models for hospital surge capacity planning: A systematic review. Disaster medicine and public health preparedness, pp.1-17.
- Fowler, Z., Moeller, E., Roa, L., Castañeda-Alcántara, I.D., Uribe-Leitz, T., Meara, J.G. and Cervantes-Trejo, A., 2020. Projected impact of COVID-19 mitigation strategies on hospital services in the Mexico City Metropolitan Area. PloS one, 15(11), p.e0241954.
- Sohrabizadeh, S., Yousefian, S., Bahramzadeh, A. and Vaziri, M.H., 2021. A systematic review of health sector responses to the coincidence of disasters and COVID-19. BMC public health, 21(1), pp.1-9.
- Kobayashi, K.J. and Alper, E.J., 2021. Coronavirus Disease 2019 Capacity Response: Where’s the Right Balance?. Medical Care, 59(5), pp.369-370.
in addition, some authors proposed some papers to make hospitals resilient to disasters that can be modified for other disasters: please see:
Aghapour, A.H., Yazdani, M., Jolai, F. and Mojtahedi, M., 2019. Capacity planning and reconfiguration for disaster-resilient health infrastructure. Journal of Building Engineering, 26, p.100853.
Keenan, J.M., 2020. COVID, resilience, and the built environment. Environment systems & decisions, p.1.
Ceferino, L., Mitrani-Reiser, J., Kiremidjian, A., Deierlein, G. and Bambarén, C., 2020. Effective plans for hospital system response to earthquake emergencies. Nature communications, 11(1), pp.1-12.
Yazdani, M., Mojtahedi, M., Loosemore, M., Sanderson, D. and Dixit, V., 2021. Hospital evacuation modelling: A critical literature review on current knowledge and research gaps. International Journal of Disaster Risk Reduction, 66, p.102627.
TohidiFar, A., Mousavi, M. and Alvanchi, A., 2021. A hybrid BIM and BN-based model to improve the resiliency of hospitals' utility systems in disasters. International Journal of Disaster Risk Reduction, 57, p.102176.
In addition, the concussion should be revised. This section (or a section that can be added before the conclusion), should provide a managerial insight.
Author Response
We really thank the reviewer for all the valuable comments that have helped us to improve our manuscript that now could be suitable for publication.
The paper is interesting and addresses an important issue. while there are some concerns that should be addressed. first of all, I do believe the title of the paper is not appropriate. As a matter of fact, there are two issues in the title, first of all, "disaster" is very confusing in the title. as a matter of fact, there is a wide range of disasters which recently some of them particularly those are related to nature, are called hazard not disaster. However, this paper is focusing on Covid and pandemic. Therefore, I suggest the author should revise the title. In addition, the author discussed Italy in the body of the paper, maybe it is better to replace "European" with something that is related to Italy.
Thank you, we changed the title in: “Public Health regulations and policies dealing with preparedness and emergency management: the experience of COVID-19 pandemic in Italy”
We added a part related to Italy in the abstract (line 26)
But one of the main issues of this paper is hospitals and their capacities planning during covid have not been discussed significantly. While I followed NEWS and know about one year ago, many European countries have many problems in capacity planning of the hospitals and providing reliable services for patients. this is a very important issue that should be discussed. please see:
- Klein, M.G., Cheng, C.J., Lii, E., Mao, K., Mesbahi, H., Zhu, T., Muckstadt, J.A. and Hupert, N., 2020. COVID-19 models for hospital surge capacity planning: A systematic review. Disaster medicine and public health preparedness, pp.1-17.
- Fowler, Z., Moeller, E., Roa, L., Castañeda-Alcántara, I.D., Uribe-Leitz, T., Meara, J.G. and Cervantes-Trejo, A., 2020. Projected impact of COVID-19 mitigation strategies on hospital services in the Mexico City Metropolitan Area. PloS one, 15(11), p.e0241954.
- Sohrabizadeh, S., Yousefian, S., Bahramzadeh, A. and Vaziri, M.H., 2021. A systematic review of health sector responses to the coincidence of disasters and COVID-19. BMC public health, 21(1), pp.1-9.
- Kobayashi, K.J. and Alper, E.J., 2021. Coronavirus Disease 2019 Capacity Response: Where’s the Right Balance?. Medical Care, 59(5), pp.369-370.
in addition, some authors proposed some papers to make hospitals resilient to disasters that can be modified for other disasters: please see:
Aghapour, A.H., Yazdani, M., Jolai, F. and Mojtahedi, M., 2019. Capacity planning and reconfiguration for disaster-resilient health infrastructure. Journal of Building Engineering, 26, p.100853.
Keenan, J.M., 2020. COVID, resilience, and the built environment. Environment systems & decisions, p.1.
Ceferino, L., Mitrani-Reiser, J., Kiremidjian, A., Deierlein, G. and Bambarén, C., 2020. Effective plans for hospital system response to earthquake emergencies. Nature communications, 11(1), pp.1-12.
Yazdani, M., Mojtahedi, M., Loosemore, M., Sanderson, D. and Dixit, V., 2021. Hospital evacuation modelling: A critical literature review on current knowledge and research gaps. International Journal of Disaster Risk Reduction, 66, p.102627.
TohidiFar, A., Mousavi, M. and Alvanchi, A., 2021. A hybrid BIM and BN-based model to improve the resiliency of hospitals' utility systems in disasters. International Journal of Disaster Risk Reduction, 57, p.102176.
Thank you for these important suggestions, we added this part (lines 262-287)
In addition, the concussion should be revised. This section (or a section that can be added before the conclusion), should provide a managerial insight.
Thank you, we revised the conclusion: lines 302-317 and 329-336
Reviewer 2 Report
Thank you for the opportunity to review the manuscript “Public Health regulations and policies dealing with preparedness and disasters’ management: a European overview”. Congratulations to the authors for their work, I found your paper a potentially very valuable resource on Health Science and therefore an interesting and relevant contribution to IJERPH.
In this commentary, the authors present an overview of the policies, regulatory frameworks and legislation on health emergency management at global and European level. Then they focus on the Italian COVID-19 pandemic as an example of management of health emergencies and they conclude by proposing some directives about the management of future emergencies.
However, in my opinion there are several aspects should be revised to improve manuscript as noted below.
SPECIFIC COMMENTS:
TITTLE
The manuscript, as the authors point out at the end of the introduction, focus on the Italian model as an example. This aspect should be reflected in the title, because it could mislead the reader.
ABSTRACT
The authors must provide some information about the Italian COVID-19 pandemic as an example of management of health emergencies.
INTRODUCTION
Correct.
LEGISLATION AND POLICIES ON HEALTH EMERGENCY MANAGEMENT
Correct.
CONCLUSION
The authors must put some perspective on the future of the legislation and policies on health emergency management in European level.
REFERENCES
Authors must review the references so that they comply with the journal's regulations
Author Response
We really thank the reviewers for all their valuable comments that have helped us to improve our manuscript that now could be suitable for publication.
Thank you for the opportunity to review the manuscript “Public Health regulations and policies dealing with preparedness and disasters’ management: a European overview”. Congratulations to the authors for their work, I found your paper a potentially very valuable resource on Health Science and therefore an interesting and relevant contribution to IJERPH.
In this commentary, the authors present an overview of the policies, regulatory frameworks and legislation on health emergency management at global and European level. Then they focus on the Italian COVID-19 pandemic as an example of management of health emergencies and they conclude by proposing some directives about the management of future emergencies.
However, in my opinion there are several aspects should be revised to improve manuscript as noted below.
SPECIFIC COMMENTS:
TITTLE
The manuscript, as the authors point out at the end of the introduction, focus on the Italian model as an example. This aspect should be reflected in the title, because it could mislead the reader.
Thank you, we changed the title in: “Public Health regulations and policies dealing with preparedness and emergency management: the experience of COVID-19 pandemic in Italy”
ABSTRACT
The authors must provide some information about the Italian COVID-19 pandemic as an example of management of health emergencies.
Thank you, we added a part related to Italy in the abstract (line 26)
INTRODUCTION
Correct.
Thank you
LEGISLATION AND POLICIES ON HEALTH EMERGENCY MANAGEMENT
Correct.
Thank you
CONCLUSION
The authors must put some perspective on the future of the legislation and policies on health emergency management in European level.
Thank you, we revised the conclusion: lines 302-317 and 329-336
REFERENCES
Authors must review the references so that they comply with the journal's regulations
Thank you
Reviewer 3 Report
This paper has much detailed content and I offer the following comments.
- I have some questions about the title. The title states European overview yet the paper includes several sections about Italy as an example. Is it European and Italy due to the length of the Italy sectiion? The paper includes content about COVID-19. If COVID-19 is the primary focus of the paper, you may want to include it in the title.
- I do not understand switching between the terminology COVID-19 and SARS-CoV-2. I recommend using one or explaining why you switch back and forth.
- The abstract lack Italy but pages 4-5 are about Italy. Italy is part of the purpose on page 2.
- The content in this paper is dense and requires concentration to read. I highly encourage making tables and/or figures to show examples or linkages of content. Tables and/or figures would make content easier to understand and assist application.
- What was learned? What should the EU do now and in future? Possibly these ideas can also be in a table.
- References appear to match the paper.
Author Response
We really thank the reviewer for all the valuable comments that have helped us to improve our manuscript that now could be suitable for publication.
This paper has much detailed content and I offer the following comments.
- I have some questions about the title. The title states European overview yet the paper includes several sections about Italy as an example. Is it European and Italy due to the length of the Italy section? The paper includes content about COVID-19. If COVID-19 is the primary focus of the paper, you may want to include it in the title.
Thank you, we changed the title in: “Public Health regulations and policies dealing with preparedness and emergency management: the experience of COVID-19 pandemic in Italy”
- I do not understand switching between the terminology COVID-19 and SARS-CoV-2. I recommend using one or explaining why you switch back and forth.
Thank you, we corrected and used only the terminology COVID-19
- The abstract lack Italy but pages 4-5 are about Italy. Italy is part of the purpose on page 2.
Thank you, we added a part related to Italy in the abstract (line 26)
- The content in this paper is dense and requires concentration to read. I highly encourage making tables and/or figures to show examples or linkages of content. Tables and/or figures would make content easier to understand and assist application. What was learned? What should the EU do now and in future? Possibly these ideas can also be in a table.
Thank you, we revised the conclusion: lines 302-317 and 329-336 and we added a table with the insights to properly manage a health emergency in Europe
- References appear to match the paper.
Thank you, the high number of references might be due to the inclusion of all the law references, such as guidelines, directives, regulations, communication from the EU Commission. Since a large part of the work focuses on legislation and policies on health emergency management, we believe it would be appropriate to retain these bibliographic references.
Round 2
Reviewer 1 Report
The authors addressed the comments.